

# NISQ algorithm for the matrix elements of a generic observable

Rebecca Erbanni[1], Kishor Bharti[2,3], Leong-Chuan Kwek[4,5,6] and Dario Poletti[1,6,7,8]

**1** Science, Mathematics and Technology Cluster, Singapore University
of Technology and Design, 8 Somapah Road, 487372 Singapore
**2** Joint Center for Quantum Information and Computer Science and Joint Quantum Institute,
NIST/University of Maryland, College Park, Maryland 20742, USA
**3** Institute of High Performance Computing (IHPC), Agency for Science, Technology and
Research (A*STAR), 1 Fusionopolis Way, #16-16 Connexis,
Singapore 138632, Republic of Singapore
**4** Centre for Quantum Technologies, National University of Singapore 117543, Singapore
**5** National Institute of Education, Nanyang Technological University,
1 Nanyang Walk, Singapore 637616
**6** MajuLab, CNRS-UNS-NUS-NTU International Joint Research Unit, UMI 3654, Singapore
**7** Engineering Product Development Pillar, Singapore University
of Technology and Design, 8 Somapah Road, 487372 Singapore
**8** The Abdus Salam International Centre for Theoretical Physics,
Strada Costiera 11, 34151 Trieste, Italy

## Abstract

The calculation of off-diagonal matrix elements has various applications in fields such as nuclear physics and quantum chemistry. In this paper, we present a noisy intermediate scale quantum algorithm for estimating the diagonal and off-diagonal matrix elements of a generic observable in the energy eigenbasis of a given Hamiltonian without explicitly preparing its eigenstates. By means of numerical simulations we show that this approach finds many of the matrix elements for the one and two qubits cases. Specifically, while in the first case, one can initialize the ansatz parameters over a broad interval, in the latter the optimization landscape can significantly slow down the speed of convergence and one should therefore be careful to restrict the initialization to a smaller range of parameters.

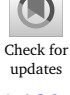
doi:[10.21468/SciPostPhys.15.4.180]()

# 1 Introduction

The landscape of quantum computing technology has shifted dramatically over the previous four decades. Once considered a theoretical endeavour, quantum computing is today a vibrant experimental pursuit. With the advent of noisy intermediate scale quantum (NISQ) devices [1,2], considerable effort has been directed into developing algorithms that can run in the presence of noise on quantum computing systems with 50–70 qubits and restricted qubit connectivity. These algorithms are known as NISQ algorithms, and common examples include the variational quantum eigensolver (VQE) [3–5] and the quantum approximate optimization algorithm (QAOA) [6,7].

There has been an explosion of recent work on NISQ algorithms to address problems such as finding ground state of Hamiltonians [3–5, 8–19] quantum simulation [20–33], combinatorial optimization [6, 7], quantum metrology [34–36], and machine learning [37–41]. Notwithstanding these research efforts, the practical application of NISQ devices is still a long way off. To attain a practical quantum advantage in the NISQ era, it is crucial to examine different algorithms. Here, we build on the works of Gerjouy et al. [42–44], which had only been implemented on classical computers, and apply them to calculating off-diagonal matrix elements of generic observables.

Many problems in nuclear physics and other sciences make extensive use of off-diagonal matrix elements. Consider the Brueckner-Bethe-Goldstone equation solution; the nuclear potential is computed from the off-diagonal matrix elements of the Brueckner reaction matrix [45]. Generalized eigenvalue problems for obtaining energy levels of time-dependent Euclidean correlators [46] in lattice quantum chromodynamics (QCD) are highly dependent on information acquired from off-diagonal elements of the matrix of correlators. The rotational-vibrational coupling in quantum chemistry considers the off-diagonal block in the matrix of kinematic coefficients [47]. Calculations of off-diagonal elements have also been used to determine the diamagnetic susceptibility and form factor of atoms such as helium [48]. Given the importance of finding the off-diagonal matrix elements of generic observables, it is pertinent to explore the potential of NISQ devices for the aforementioned task. Recent work has shown that the complexity of the measurement circuit in NISQ devices can be significantly reduced, thus significantly improving their performance [49–56], and that one could tap on

them and [10] to evaluate off-diagonal elements with an extended swap test circuit with one ancilla qubit.

In this paper, we provide an NISQ algorithm for hybrid classical-quantum computation of matrix elements (both diagonal and off-diagonal) of a given observable $W$ in the energy eigenbasis. These are typically complex numbers and one cannot simply minimize them as in standard VQE. However, variational approaches have been used for decades on classical computers, and we can build on this experience and test the performance of more general variational approaches, as the ones put forward in Refs. [42–44] on NISQ devices. Our algorithm thus uses a variational function which, at its equilibrium points, returns the values we aim to compute. This is possible thanks to the introduction of purposefully built Lagrange multipliers to encode the constraints for the underlying problem. For our model problems, we discuss approaches for both exact and iterative evaluation of Lagrange multipliers. Various numerical simulations show that, for a single-qubit problem, our approach can find all the matrix elements even when one initializes randomly the trial functions over a very broad range of parameters. For two-qubit problems one may need to prepare the state closer to an eigenstate, although further improvements are still to be explored.

We would like to point out that a quantum variational approach for calculating matrix elements was proposed recently in [57], which relied on the preparation of the energy eigenstates corresponding to which the overlap for the given observable is to be calculated. Unlike this work, our approach does not involve the preparation of the energy eigenstates, hence avoiding the accompanying quantum resource requirements. Although our choice of ansatz may seem to resemble existing work in the literature [8, 12, 18], none of these results works for the off-diagonal matrix elements of a generic observable. Moreover, our approach is fundamentally different and uses Lagrange multipliers to encode the constraints for the underlying problem into the refined objective. Using Lagrange multipliers, it is possible to convert a constrained optimization problem into an unconstrained optimization problem, as the constraints are incorporated as a component of the objective. This approach enables the utilization of techniques for unconstrained optimization, such as those based on the derivative test.

In our algorithm, the number of overlaps to be evaluated scales efficiently with the number of basis states utilized in designing the ansatz. Furthermore, it is worth noting that these overlaps need to be computed only once, after which they are multiplied by the coefficients obtained from the optimization process in the iterative algorithm.

## 2  The classical variational algorithms for off-diagonal elements

Given a system with Hamiltonian $H$ with different eigenenergies $E_i$ and the corresponding eigenfunctions $|\phi_i\rangle$ and an observable $W$, what we aim to find are the elements $F_{i,j} = \langle \phi_i | W | \phi_j \rangle$.

One natural approach would be to find the different eigenfunctions of the Hamiltonian and then evaluate the elements, including the off-diagonal ones. However, one may not need to do this. In [42, 43] the authors showed a variational approach to find such elements which we summarize in the following. First, we can use a variational ansatz for the eigenfunctions $|\phi_i\rangle \approx |\phi_{i,t}(\vec{\eta}_i)\rangle$ parametrized by the parameters $\vec{\eta}_i$. We can then write a variational function $F_{i,j}^v$ that has a zero derivative to $\vec{\eta}_i$ when $F_{i,j}^v = F_{i,j}$. Such an approach can readily give both diagonal and off-diagonal elements. In the following we consider a normalized parametrization of the trial eigenfunctions $|\phi_{i,t}(\vec{\eta}_i)\rangle$, and an extension to the case of non normalized trial eigenfunctions is discussed in the App.C.

We first note that, for a given Hermitian matrix $W$, we can always write $W_R = (W + W^T)/2$ and $W_I = (W - W^T)/2$, where $A^T$ is the transposition of $A$, respectively for the real components and for the imaginary ones. We can then consider the case for which the observable $W$ is only

real or only imaginary, and for this we build a variational function $F_\nu$ which is given by $F_{i,j}^\nu$ plus Lagrangian multipliers multiplied by the constraints. These multipliers and constraints are built such that $F_\nu = F_{i,j}^\nu$ at its equilibrium points. More specifically, the variational function $F_{i,j}^\nu$ is given by

$$
\begin{aligned}
F_\nu = {} & \langle \phi_{i,t} | W | \phi_{j,t} \rangle \\
& + \langle L_{i,a} | (H - E_i) | \phi_{i,t} \rangle + \langle \phi_{i,t} | (H - E_i)^\dagger | L_{i,b} \rangle \\
& + \langle L_{j,a} | (H - E_j) | \phi_{j,t} \rangle + \langle \phi_{j,t} | (H - E_j)^\dagger | L_{j,b} \rangle \\
& + \lambda \left[ \langle \phi_{i,t} | W | \phi_{j,t} \rangle \mp \langle \phi_{j,t} | W | \phi_{i,t} \rangle \right],
\end{aligned}
\tag{1}
$$

where, in the last line, one uses the sign $-$ or $+$ depending on whether $W$ is real or imaginary respectively. In Eq.(1), the first term is the expectation value we aim to compute, while the others are the constraints with their corresponding Lagrange multiplier. The second and third lines set the constraint that $|\phi_{i,t}\rangle$ and $|\phi_{j,t}\rangle$ are eigenstates, while the last line is to ensure that the function estimates only the real or the imaginary part. Note that one can expand the variational algorithm to the case of non-normalized wavefunctions simply by adding new constraints on the normalization of $|\phi_{i,t}\rangle$ to Eq.(1) as shown in [42, 43]. In Eq.(1) we have used the Lagrange multiplier vectors $|L_{i,\nu}\rangle$ (which, generally, are not normalized) and the scalar $\lambda$. The expressions to evaluate the Lagrange multipliers $L_i$ and $\lambda$, is obtained by expanding Eq. (1) to first order, and sets all the terms to zero. Details of such computations are found in App.A. For instance, we can take a small variation $|\delta\phi_i\rangle$ to the exact eigenfunctions $|\phi_i^{ex}\rangle$, which gives $|\phi_{i,t}\rangle = |\phi_i^{ex}\rangle + |\delta\phi_i\rangle$, and setting the first order corrections to the exact result of the function $F_{i,j}^\nu$ to zero, we obtain

$$
\lambda = -1, \tag{2}
$$

$$
(H - E_i)|L_{i,\nu}\rangle = -\xi_{R,I}^\nu W_{R/I} |\phi_j\rangle. \tag{3}
$$

with $\xi_{R,I}^\nu = \pm 1$. More precisely, for the real case $\xi_R^\nu = 1$ whether $\nu = a, b$ while for the imaginary case $\xi_R^a = 1$ and $\xi_R^b = -1$. What is important to state, though, is that in principle we do not know the value $E_i$ and thus this will be approximate by the expectation value of the Hamiltonian for that wave function, i.e. $E_i \approx \langle \phi_i | H | \phi_i \rangle$.

At this point we should solve Eq.(3) for $|L_{i,\nu}\rangle$, but this is not straightforward because $H - \langle \phi_i | H | \phi_i \rangle$ is not invertible. For the purpose of computing these Lagrange multipliers we thus use a modified Hamiltonian

$$
H_{mod,i} = H - \frac{H |\phi_i\rangle \langle \phi_i| H}{\langle \phi_i | H | \phi_i \rangle}, \tag{4}
$$

such that the matrix $H_{mod,i} - \langle \phi_i | H | \phi_i \rangle$ is not singular.

When it is difficult to evaluate the inverse of $H_{mod,i} - \langle \phi_i | H | \phi_i \rangle$ exactly, it is also possible to implement an iterative approach. As shown in [43], we can find $|L_{i,\nu}\rangle$ by minimizing

$$
M(|L_{i,\nu}\rangle) = \langle L_{i,\nu} | \left( H_{mod,i} - \langle \phi_i | H | \phi_i \rangle \right) | L_{i,\nu} \rangle + \langle \phi_j | W_R | L_{i,\nu} \rangle, \tag{5}
$$

for the real case, and

$$
M(|L_{i,\nu}\rangle) = \langle L_{i,\nu} | \left( H_{mod,i} - \langle \phi_i | H | \phi_i \rangle \right) | L_{i,\nu} \rangle, \tag{6}
$$

for the imaginary one. One thus ends up with two variational principles, one for the derivation of the Lagrange multipliers $|L_{i,\nu}\rangle$ and one for the derivation of matrix elements $\langle \phi_{i,t} | W | \phi_{j,t} \rangle$. More details on the derivation of Eq. 5 and 6 can be found in App.B.

## 2.1 Scaling analysis for the overlap calculation

We consider Hamiltonians $H$ and observables $W$ which can be written as linear combination of poly$(n)$ unitaries where $n$ is the number of qubits over which the Hamiltonian is defined. A typical example is a local spin Hamiltonian which can be written as a sum of polynomially many $n$-qubit Pauli matrices. Furthermore, we consider a number of ansatz states and Lagrange multipliers that scales polynomially with the system size (number of qubits) as this is often sufficient to obtain accurate results, e.g. using a Krylov basis [18, 58–60].

We thus highlight that there are three types of overlaps that need to be computed for the successful implementation of our NISQ algorithm,

- $\langle \phi_i | H | \phi_i \rangle$: Since the number of terms in the Hamiltonian and the number of ansatz states $|\phi_i\rangle$ are polynomially many in number of qubits, the overlaps $\langle \phi_i | H | \phi_i \rangle$ can be calculated efficiently [61].

- $\langle L_{i,\nu} | H_{mod,i} | L_{i,\nu} \rangle$: the estimation of these overlaps requires the calculation of the terms of the form $\langle \phi_i | H | \phi_i \rangle$, $\langle L_{i,\nu} | H | L_{i,\nu} \rangle$ and $\langle L_{i,\nu} | H | \phi_i \rangle$. Since the number of ansatz states and Lagrange multipliers scale polynomially with the system size, also the aforementioned overlaps can be evaluated efficiently.

- $\langle \phi_j | W_R | L_{i,\nu} \rangle$: Since the operator $W$ can be expressed as a linear combination of polynomially many unitaries, it is easy to see that also the overlaps $\langle \phi_j | W_R | L_{i,\nu} \rangle$ can be calculated efficiently.

The number of overlaps to evaluate scales as $\mathcal{O}(n^2 + nm)$ where $n$ is the number of basis states of $\left| \phi_{i,t} \right\rangle$ and $m$ that of $\left| L_{i,\nu} \right\rangle$. Here, $n^2$ comes from terms such as $\left\langle \phi_{i,t} \right| W \left| \phi_{j,t} \right\rangle$, while $nm$ comes from terms of type $\left\langle L_{i,a} \right| (H - E_i) \left| \phi_{i,t} \right\rangle$. As previously explained, all these overlaps only have to be computed once and are then multiplied by the coefficients resulting from the optimization process in the iterative algorithm. It should be noted, however, that the accuracy resulting from the number of queries to the quantum computer used to evaluate the overlaps depends on many factors including the number of basis elements considered, the size of the system and both the Hamiltonian and observable under investigation.

## 3 Results

We will now show examples which elucidate the effectiveness of an hybrid classical-quantum implementation of this variational approach. We will consider both a single and a two-qubit Hamiltonian and we will use both the exact and the iterative approaches to get the Lagrange multiplier.

## 3.1 Models

We consider two scenarios. The first scenario is a two level system with Hamiltonian

$$H_1 = X, \tag{7}$$

and $W_1 = H_d W_1^D H_d$ where

$$W_1^D = \begin{pmatrix} 5 & 2 - 2j \\ 2 + 2j & 3 \end{pmatrix}. \tag{8}$$

$W_1^D$ is thus the matrix in the energy eigenbasis, as $H_d$ diagonalizes the Hamiltonian $H_1$.

In the second case we consider the following $2-$ qubit Hamiltonian

$$H_2 = 2X \otimes \mathbb{I} + \mathbb{I} \otimes X + 2Z \otimes X, \tag{9}$$

while we take an $W_2^D$ (therefore in the energy eigenbasis) as the following Hermitian complex matrix

$$W_2^D = \begin{pmatrix} 1 & 3+1j & 5-3j & 13+8j \\ 3-1j & 4 & 20+5j & 25+10j \\ 5+3j & 20-5j & 7 & 6-15j \\ 13-8j & 25-10j & 6+15j & 10 \end{pmatrix}. \tag{10}$$

The matrix $W_2$ in the computational basis, which we use in the computations, can be obtained from $W_2^D$ and the eigenvalues of $H_2$.

## 3.2 Implementation for hybrid classical-quantum computation

We test the usefulness of the variational principle from Eq.(1) on a hybrid classical-quantum algorithm. Since the solutions can be found where the derivatives are zero, we use a classical optimization algorithm which performs a gradient descent after having evaluated the gradients of $F_{i,j}^{\nu}$ over the parameters $\vec{\eta}_i$. The quantum part of the algorithm helps with the evaluation of the gradients. In practice, one can evaluate all the relevant overlaps once, and then use such knowledge to evaluate the gradients for any given value of the $\vec{\eta}_i$. To evaluate the overlaps we write the states $\left|\phi_{i,t}\right\rangle$ as

$$\left|\phi_{i,t}(\theta)\right\rangle = \cos(\theta_i)\left|0\right\rangle + \sin(\theta_i)\left|1\right\rangle. \tag{11}$$

for the one-qubit case, and for the two-qubit case we consider the parametrization

$$\begin{aligned} \left|\phi_{i,t}(\alpha_i, \beta_i, \gamma_i)\right\rangle = {}& \cos(\alpha_i)\left|00\right\rangle + \sin(\alpha_i)\cos(\beta_i)\left|01\right\rangle \\ & + \sin(\alpha)\sin(\beta_i)\cos(\gamma_i)\left|10\right\rangle + \sin(\alpha_i)\sin(\beta_i)\sin(\gamma_i)\left|11\right\rangle. \end{aligned} \tag{12}$$

Note that here we only consider real trial functions, which is sufficient for our examples, and a generalization to complex ones is straightforward. Furthermore, we currently use a variational representation of the eigenfunctions which scales linearly with the size of the Hilbert space. In practice, for systems with large Hilbert space, for which a quantum computer would come in handy, one would have to resort to a much smaller parameter space, for example using a limited Krylov basis [62, 63]. For the Lagrange multipliers $\left|L_{i,\nu}\right\rangle$ we use an unnormalized ansatz of the form

$$\left|L_{i,\nu}\right\rangle = c_i\left|0\right\rangle + d_i\left|1\right\rangle, \tag{13}$$

for the one qubit case and

$$\left|L_{i,\nu}\right\rangle = c_i\left|00\right\rangle + d_i\left|01\right\rangle + e_i\left|10\right\rangle + f_i\left|11\right\rangle, \tag{14}$$

for the qubit case, where the parameters of Eq.(13,14) are real numbers.

All quantum computations are implemented on the IBM Belem QPU simulator [64] using error mitigation. The overlaps are evaluated by averaging 50 estimates of the overlaps each done with 1000 shots.

## 3.3 Single qubit using an exact evaluation of the Lagrange multipliers

We now consider the model with a single qubit, with Hamiltonian $H_1$ from Eq.(7) and the matrix $W_1$ derived from the matrix in the energy eigenbasis Eq.(8). We use a classical optimization algorithm by deriving analytically the derivatives of Eq. (1) with respect to the angles



Figure 1: Plots for the real and imaginary parts of the matrix elements versus iterations of the classical optimization algorithm, through exact calculation of $\left|L_{1,v}\right\rangle$ and $\left|L_{2,v}\right\rangle$ in the 1-qubit case and with error mitigation. Panels (a,c,e) are the results for the real part of $W_1$ in the energy eigenbasis, while (b,d,f) for the imaginary part. Panels (a,b) show the matrix elements while panels (c) to (f) show the angles $\theta_i$ of the trial eigenfunctions. Each panel shows the results from a total of 150 runs, where the angles have been randomly initialized between 0 and $2\pi$. In all panels, the lines represent the medians of the runs converging to a specific value and the error intervals include 92% of the corresponding runs. In panels (c) and (e), the angles have been mapped to be $\in (-\frac{\pi}{2}, \frac{\pi}{2})$ in order to fix the global phase of the $\phi_i$, as detailed in Section 3.3.

of Eqs. (11) and the parameters of Eq. (13); the expectation values and gradients are evaluated using overlaps computed on the IBM Belem quantum processor's simulator. In Fig. 1 we show the estimated value of the element of $W_{R/I}$ versus number of iterations of the classical optimization procedure, panels (a) and (b), and the values of the angles $\alpha_i$ which parametrize the trial eigenfunctions, panels (c) to (f). In the left panels, (a, c, e), we consider the real part of the observable $W$, i.e. $W_R$, while in the right panels (b, d, f), the imaginary part, $W_I$. In each of the two cases we show the results from 150 runs of the protocol with initial angles for the two eigenfunctions $\alpha_1$ and $\alpha_2$ chosen uniformly between $-\pi$ and $\pi$. The line in each of the panels is obtained by the median value of the expectation value or angle between 50 runs which end in the same vicinity. The colored background reflects the value taken by the middle 92% of the corresponding realizations. In panels (a) and (b) we observe that while in the initial steps the angles cover the all range from $-\pi$ to $\pi$, and the expectation values take a very large range of possible values, within 20 iterations the prediction of the matrix elements is very accurate, both for the real, panel (a), and imaginary part, panel (b). We note that the accuracy in the angles may not be as good as that on the matrix elements. This is one advantage of using this variational approach which is tailored to give the matrix elements directly. As a technical, but important, detail: for $W_R$, we had to fix the global phase of the trial eigenfunctions, otherwise the expectation value may show the wrong sign. For this reason, when plotting the angles and evaluating the corresponding matrix elements, we mapped them between the angles $-\pi/2$ and $\pi/2$ so that the cosine of the angle would be positive. Such procedure is not needed for the imaginary elements of $W$ because they have boths signs for each value since the matrix is Hermitian, and for this reason the plotted range of angles in panels (d) and (f) is between 0 and $2\pi$.

One could also think of variationally computing the ground and excited states by extending the well-known VQE to higher-energy states with the addition of a constraint on the orthogonality of the $|\phi_i\rangle$ [65], but in order to compute the $n-$th eigenstate, this method requires to also compute all $(n-1)$ previous states. This approach can also work, however in a different way from the one we use in which we can randomly initialize the system parameters and explore the expectation values that emerge. We could also fix $E_i$ and $E_j$ with $E_i < E_j$, and compute the ground state of a new Hamiltonian $(H - E_j)^2$, and then evaluate the off-diagonal matrix elements of $W$. However, this approach requires a highly non-local Hamiltonian.

Lastly, when computing the quantity $\langle \phi_i | W | \phi_j \rangle$ one could use indirect measurement methods e.g. Hadamard test, that, with the help of an additional ancilla, can return the real and imaginary parts of $\langle \phi_i | W | \phi_j \rangle$, see [10]. This comes at the expense though, of having to implement the controlled$-U$ operation, which is challenging for NISQ devices [66]. Instead, we have only considered unitaries that are made of linear combinations of Pauli matrices, since their expectation values can be computed efficiently, see Sec. 2.1 and Refs. [49–56].

## 4 Analysis of errors for the single-qubit case

For the single qubit case, both for the real and imaginary part we expect to obtain three different numbers from the variational approach. In Fig.2 we consider, in each panel, the error from each of these six possible values. More specifically, we consider the real values 2, 5 and 3 in panels (a), (c) and (e) and the values from the imaginary part 2i, $-2i$ and 0 in panels (b), (d) and (f). In each panel we show the median of the 50 runs approaching that value for three different cases: completely classical simulations (orange lines), hybrid classical-quantum simulations without error mitigation (green lines) and with error mitigation (blue lines). Also in this case the shadowing represent the 92% confidence interval of the corresponding runs (i.e. between the 4−th and the 96−th percentile). We observe that only the fully classical

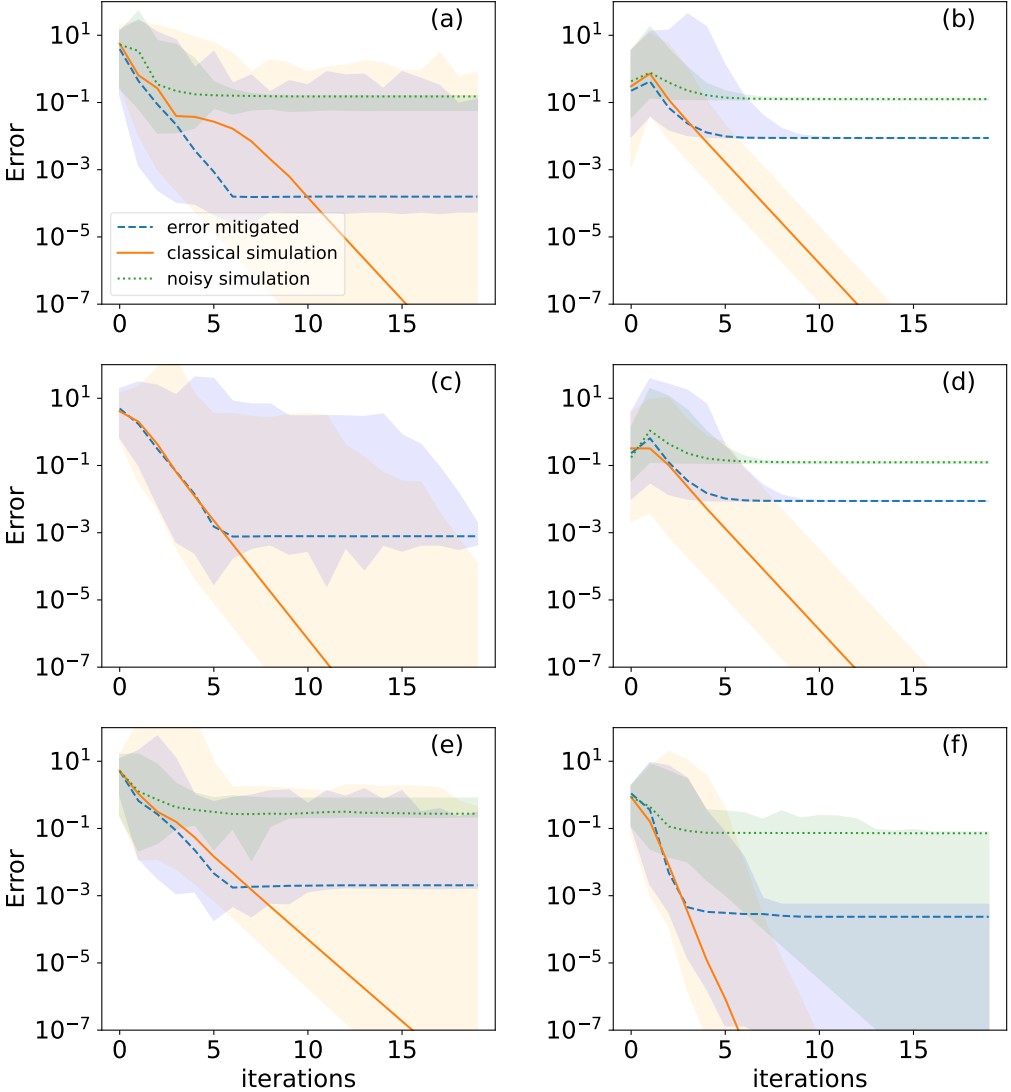

Figure 2: Absolute value of the errors for each matrix element along 20 iterations, for classical, noisy and error mitigated simulations, along with the 92% confidence level of the points. Both columns refer to experiments where the Lagrange multipliers $\left|L_{1,\nu}\right\rangle$ and $\left|L_{2,\nu}\right\rangle$ are computed exactly as discussed in Sec. 2. Each panel (a-f) corresponds respectively to the matrix elements (in the energy eigenbasis) 2, 2i, 5, −2i, 3 and 0. The orange continuous line corresponds to classical simulations, the blue dashed line to hybrid classical-quantum simulations with error mitigation, and the green dotted line to hybrid classical-quantum simulations with no error mitigation. The presence of only the blue dashed and orange continuous lines in panel (c) imply that the value from the hybrid computation without error correction, represented in other panels by the green dotted line, has not converged to a value close enough to 5.

approach is able to reach very small errors and continuously improves as the number of iterations of the optimization routine increase. At the same time, also for the classical approach the process shows a non-negligible error bar. For the hybrid approach, instead, we observe that the error reaches a plateau after about 10 iterations. This error is reduced when employing error-mitigation techniques. We thus associate this performance to an erroneous evaluation of the overlaps for the gradients. In some cases, as in panel (c) for the value 5, the errors are so



Figure 3: Exactly the same description as Fig. 1 with the only difference that in the optimization procedure we have used approximated Lagrange multipliers from minimization of Eq.(5) or Eq.(6).

important that the hybrid classical-quantum algorithm is not able to converge to this solution when one does not implement error mitigation.

## 4.1 Single qubit using an approximate evaluation of the Lagrange multipliers

Fig. 3 is completely analogous to Fig. 1, and thus it shows results for real and imaginary elements of $W$ and the corresponding angles versus the number of iterations. However, in this case we used the iterative algorithm from finding the minima of Eq.(5) and Eq.(6). A detailed description of this code can be found in App. D. Since in this case the Lagrange multipliers $|L_{i,\nu}\rangle$ are not exact, the optimization is not as accurate. Hence while the medians of the expectation

values and of the angles approach the exact values, the error bars, here also indicated by the 92% confidence level of the runs ending close to one solution, are larger.

## 5   Two qubits

To understand how this variational approach would perform on larger systems, we now consider a two-qubit case. We will use the Hamiltonian $H_2$ from Eq.(9) and the observable $W_2$ which in the eigenbasis of the Hamiltonian has the values $W_2^D$ from Eq.(10). The trial eigenfunctions are parametrized as shown in Eq.(12). We consider 300 different initializations and we show, in Fig. 4 a density plot of the resulting matrix elements versus the number of iterations. By density plot, we mean that for each of the 20 iterations, we consider a vertical range and divide it into small bins of length 0.2 and count how many points fall into those intervals. The initial conditions are prepared near the exact angles for the solutions of the eigenfunctions in Eq.(12) with an error $\pm 0.15$ from the exact $\alpha_i$, $\beta_i$ and $\gamma_i$. Furthermore, noise from the quantum machine has not been considered in these two-qubit calculations, and only statistical errors, i.e. shot-noise, are considered.

From Fig. 4, we find that some matrix elements are much more stable than others. For instance the elements 20, 25 and 6 converge in few iterations and they have a large probability of appearing. Other values, like 13, are unstable and they do not appear unless one chooses initial conditions for the parameters of the trial eigenfunctions very close to the exact ones. This is true also for completely classical simulations, hinting at the fact that the landscape of this optimization problem is particularly difficult, and better classical optimization routines should be used. What is possibly more striking is that there seems to be converged results to values which do not belong to $W_2$, as for instance the value $\approx 30$. Computing for longer times we observe that this value drifts, indicating that it is not converged, and hence it is not a value predicted by the model. From this we deduce that this method can give a good number of matrix elements, and depending on the amount of noise in the machine, they can be fairly accurate, but it may also not find some values and possibly return, after limited iterations, a few values which are not correct.

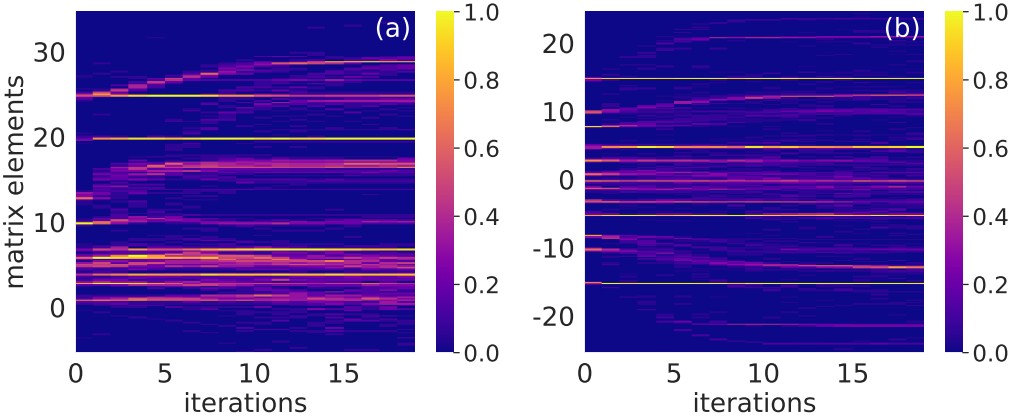

Figure 4: Heatmaps for the two-qubit case and for the real (a) and imaginary (b) elements of $W_2$ in energy eigenbasis, through exact calculation of the Lagrange multipliers. Both subplots show the convergence to the matrix elements for 300 runs of 20 iterations each, initialized in intervals of radius 0.15 of the optimal angles.

# 6 Conclusions

We have considered a variational approach for hybrid classical-quantum computation of the matrix elements of an observable $W$ in the energy eigenbasis. We have implemented the method with one qubit to learn some of its basic features, and then we have proceeded to two qubits to study how the method performs when the size of the systems studied increases. We have found that the method performs well in the case of one qubit, finding real and imaginary parts of all matrix elements even when one initializes randomly the trial functions over a very broad range of parameters. For two qubits, we have provided evidence that the algorithm can still work (in the sense that it can find the target values) but, in general, it requires an initialization of the variational parameters much closer to the exact value. As future work, we may investigate the role of different optimization routines in the classical portion of the hybrid computation. Overall, the performance of the method is limited by the errors in the estimation of the gradients and overlaps from the quantum computer, which can be tamed with less noisy devices and further sampling. We highlight that one can use a much smaller basis, which represent states closer to those of interest, thus limiting the emergence of spurious incorrect results. In this case the optimization procedure will also be better controlled. Another way to improve the performance is to re-initialize the angles from the results after a certain number of iterations. Here we used normalized trial functions, but the method can be generalized, as we show in App.C, to non-normalized trial states.

# Acknowledgments

**Funding information** K.B. acknowledges funding by U.S. Department of Energy Award No. DE-SC0019449, DoE ASCR Accelerated Research in Quantum Computing program (award No. DE-SC0020312), DoE QSA, NSF QLCI (award No. OMA-2120757), NSF PFCQC program, DoE ASCR Quantum Testbed Pathfinder program (award No. DE-SC0019040), AFOSR, ARO MURI, AFOSR MURI, and DARPA SAVaNT ADVENT. KLC acknowledges support from the Ministry of Education and the National Research Foundation, Singapore. D.P. acknowledges support from NRF-ISF grant NRF2020-NRF-ISF004-3528.

# Appendix

# A Variational principle

The variational principle is best gleaned from examples. Here, we restrict ourselves to the determination of off-diagonal matrix elements in quantum mechanics. In order to do so, we write the expression of the variational principle starting from the trial objective and the constraints, i.e.

$$F_v = \left\langle \phi_{i,t} \middle| W \middle| \phi_{j,t} \right\rangle, \tag{A.1}$$

and

$$B_1 = (H - E_{i,j}) \middle| \phi_{i,j} \right\rangle = 0 \text{ and } B_1^\dagger = \left\langle \phi_{i,j} \middle| (H - E_{i,j}) = 0. \tag{A.2}$$

The final expression is obtained by promoting the constraints to the objective by multiplication by some Lagrange multipliers $L$ and $\lambda$. The goal is then to find expressions for $L$ and $\lambda$ either exactly or iteratively, and to check which ones are best suited for the variational principle.

Concretely, the trial quantities of interest and the trial Lagrange multipliers are defined as

$$\left|\phi_{i/j,t}\right\rangle = \left|\phi_{i/j}\right\rangle + \left|\delta\phi_{i/j}\right\rangle, \tag{A.3}$$

$$\lambda_t = \lambda + \delta\lambda, \tag{A.4}$$

and

$$\left|L_{i/j,t}\right\rangle = \left|L_{i/j}\right\rangle + \left|\delta L_{i/j}\right\rangle. \tag{A.5}$$

From this, we write the variational form of F, $F_v$, as

$$
\begin{aligned}
F_v = & \left\langle\phi_{i,t}\right|W\left|\phi_{j,t}\right\rangle \\
& + \left\langle L_{i,a,t}\right|(H-E_i)\left|\phi_{i,t}\right\rangle + \left\langle\phi_{i,t}\right|(H-E_i)^\dagger\left|L_{i,b,t}\right\rangle \\
& + \left\langle L_{j,a,t}\right|(H-E_j)\left|\phi_{j,t}\right\rangle + \left\langle\phi_{j,t}\right|(H-E_j)^\dagger\left|L_{j,b,t}\right\rangle \\
& + \lambda\left[\left\langle\phi_{i,t}\right|W\left|\phi_{j,t}\right\rangle \mp \left\langle\phi_{j,t}\right|W\left|\phi_{i,t}\right\rangle\right],
\end{aligned}
\tag{A.6}
$$

and by replacing the trial quantities with equations A.3-A.5, one can get the error $\delta F_v$ as

$$
\begin{aligned}
\delta F_v = & F_v - \left\langle\phi_{i,t}\right|W\left|\phi_{j,t}\right\rangle \\
= & (\langle\phi_i| + \langle\delta\phi_i|)W(|\phi_j\rangle + |\delta\phi_j\rangle) - \langle\phi_i|W|\phi_j\rangle + (\langle L_{i,a}| + \langle\delta L_{i,a}|)[(H-E_i)(|\phi_i\rangle + |\delta\phi_i\rangle)] \\
& + [(\langle\phi_i| + \langle\delta\phi_i|)(H-E_i)^\dagger](|\delta L_{i,b}\rangle + |L_{i,b}\rangle) + (\langle L_{j,a}| + \langle\delta L_{j,a}|)[(H-E_j)(|\phi_j\rangle + |\delta\phi_j\rangle)] \\
& + [(\langle\phi_j| + \langle\delta\phi_j|)(H-E_j)^\dagger](|\delta L_{j,b}\rangle + |L_{j,b}\rangle) + (\lambda + \delta\lambda)((\langle\phi_i| + \langle\delta\phi_i|)W(|\phi_j\rangle + |\delta\phi_j\rangle) \\
& \mp (\langle\phi_j| + \langle\delta\phi_j|)W(|\phi_i\rangle + |\delta\phi_i\rangle)) = 0.
\end{aligned}
$$

By using constraints A.2 and A.3, discarding terms of second order and putting equal to 0 the coefficients of $|\delta\phi\rangle$ and $\langle\delta\phi|$, $\delta F_v$ vanishes for all allowed $|\delta\phi\rangle$ and $\langle\delta\phi|$, and we get, for the real case

$$\langle\delta\phi_i| \to (H-E_i)\left|L_{i,b}\right\rangle = -(\lambda+1)W\left|\phi_j\right\rangle, \tag{A.7}$$

$$\langle\delta\phi_j| \to (H-E_j)\left|L_{j,b}\right\rangle = \lambda W\left|\phi_i\right\rangle, \tag{A.8}$$

$$|\delta\phi_i\rangle \to \left\langle L_{i,a}\right|(H-E_i) = \lambda\left\langle\phi_j\right|W, \tag{A.9}$$

and

$$\left|\delta\phi_j\right\rangle \to \left\langle L_{j,a}\right|(H-E_j) = -(\lambda+1)\left\langle\phi_i\right|W. \tag{A.10}$$

Multiplying on the left of equation A.7 by $\langle\phi_i|$ and on the left of equation A.8 by $\langle\phi_j|$, and applying constraint A.2, we get $\lambda = -1/2$. $|L\rangle$ can be obtained by $|L\rangle = c_1|\phi\rangle$ and made unique by $\langle\phi|L\rangle = c_2 = 1$. We can see that for the real case $\left|L_{i,a}\right\rangle = \left|L_{i,b}\right\rangle$ and $\left|L_{j,a}\right\rangle = \left|L_{j,b}\right\rangle$.

For pure imaginary matrix elements, the procedure is the same, and we get

$$\langle\delta\phi_i| \to (H-E_i)\left|L_{i,b}\right\rangle = -(\lambda+1)W\left|\phi_j\right\rangle, \tag{A.11}$$

$$\langle\delta\phi_j| \to (H-E_j)\left|L_{j,b}\right\rangle = -\lambda W\left|\phi_i\right\rangle, \tag{A.12}$$

$$|\delta\phi_i\rangle \to \left\langle L_{i,a}\right|(H-E_i) = -\lambda\left\langle\phi_j\right|W, \tag{A.13}$$

and

$$\left|\delta\phi_j\right\rangle \to \left\langle L_{j,a}\right|(H-E_j) = -(\lambda+1)\left\langle\phi_i\right|W. \tag{A.14}$$

Again, $\lambda = -1/2$ but this time $\left|L_{i,a}\right\rangle = -\left|L_{i,b}\right\rangle$ and $\left|L_{j,a}\right\rangle = -\left|L_{j,b}\right\rangle$.

## B  Variational principle for $L_{i,v}$

The next step is to derive an expression, exact or approximated, for computing the Lagrange multiplier $|L_{i,v}\rangle$. This could in theory be done exactly by inverting $(H-E_i)$ in A.7. In practice, a near-singularity problem arises because the operator $H-E_i$, where $E_i$ is the eigenstate energy, has a zero eigenvalue and therefore cannot be inverted; this can be solved by replacing said operator by a shifted one that does not have a zero eigenvalue as discussed in the main text. In addition, in order to get an extremum principle that would give us an approximation to $|L_{i,v}\rangle$ without inverting any matrix, this operator must be positive definite.

An operator that satisfies this condition is $H_{mod,i} - E_{i,t} = H - \frac{H P_{i,t} H}{E_{i,t}} - E_{i,t}$ where $P_{i,t}$ is the trial projection operator to the eigenstate $P_{i,t} = |\phi_{i,t}\rangle\langle\phi_{i,t}|$ and $E_{i,t}$ is the trial eigenstate energy $E_{i,t} = \langle\phi_{i,t}|H|\phi_{i,t}\rangle$. The idea is to find another variational principle for $|L_t\rangle$ (where we omit the indexes $i$ and $v$ to lighten the notation) that would replace Eq. (A.11-A.14) , and that could then be used in the variational principle of Eq. (A.6) to find better approximations of the trial functions. This variational principle for $|L_t\rangle$ is of the form

$$M(|X_{tt}\rangle) = \langle X_{tt}|A|X_{tt}\rangle - \langle X_{tt}|q_t\rangle - \langle q_t|X_{tt}\rangle , \tag{B.1}$$

where A will be the shifted non-negative operator, $|X_t\rangle$ is the trial Lagrange multiplier $|L_{tt}\rangle$, $|q_t\rangle = A|X_t\rangle$ a known function and we write $|X_{tt}\rangle = |X_t\rangle + |\delta X_t\rangle$. By using this last equivalence, eq B.1 becomes

$$M(|X_t\rangle + |\delta X_t\rangle) = M(|X_t\rangle) + \langle\delta X_t|A|\delta X_t\rangle , \tag{B.2}$$

where the quadratic term on the far right is strictly convex since A is positive definite. This expression has a minimum for $|X_{tt}\rangle = |X_t\rangle$, i.e. $|L_{tt}\rangle = |L_t\rangle$.
Setting $|q_t\rangle = A|X_t\rangle = (H_{mod,i} - E_{i,t})|L_t\rangle = [(\langle\phi_{i,t}|W|\phi_{i,t}\rangle) - E_{i,t}c_{i,t}]|\phi_{i,t}\rangle - W|\phi_{i,t}\rangle$, the variational principle for $|L_t\rangle$ becomes

$$M(L_{tt}) = \langle L_{tt}|(H_{mod,i} - E_{i,t})|L_{tt}\rangle - \langle L_{tt}|q_t\rangle - \langle q_t|L_{tt}\rangle , \tag{B.3}$$

and with $|L_{tt}\rangle = |L_t\rangle + |\delta L_t\rangle$,

$$M(|L_t\rangle + |\delta L_t\rangle) = M_{11}(|L_t\rangle) + \langle\delta L_t|(H_{mod,i} - E_{i,t})|\delta L_t\rangle , \tag{B.4}$$

which has its minimum at $|L_{tt}\rangle = |L_t\rangle$.
For the real case, eq B.3 becomes:

$$M(L_{i/j}) = \langle L_{i/j}|(H_{mod,i/j} - E_{i/j})|L_{i/j}\rangle + \langle\phi_{j/i}|W|L_{i/j}\rangle , \tag{B.5}$$

while for the imaginary one, it is

$$M(L_{i,b/j,b}) = \langle L_{i,b/j,b}|(H_{mod,i/j} - E_{i/j})|L_{i,b/j,b}\rangle . \tag{B.6}$$

## C  Variational principle for non normalized states

The variational principle for non-normalized wave functions requires the introduction of the following additional constraints

$$\langle\phi_{i/j}|\phi_{i/j}\rangle - 1 = 0 , \tag{C.1}$$

and the respective Lagrange multipliers $\lambda_{i/j}$. The variational principle then becomes

$$
\begin{aligned}
F_v = {}& \langle\phi_{i,t}|W|\phi_{j,t}\rangle \\
& + \langle L_{i,a,t}|(H-E_i)|\phi_{i,t}\rangle + \langle\phi_{i,t}|(H-E_i)^\dagger|L_{i,b,t}\rangle \\
& + \langle L_{j,a,t}|(H-E_j)|\phi_{j,t}\rangle + \langle\phi_{j,t}|(H-E_j)^\dagger|L_{j,b,t}\rangle \\
& + \lambda_i(\langle\phi_{i,t}|\phi_{i,t}\rangle-1) + \lambda_j(\langle\phi_{j,t}|\phi_{j,t}\rangle-1) \\
& + \lambda\left[\langle\phi_{i,t}|W|\phi_{j,t}\rangle \mp \langle\phi_{j,t}|W|\phi_{i,t}\rangle\right],
\end{aligned}
\tag{C.2}
$$

and we get the following equations for the real case

$$
\langle\delta\phi_i|: (H-E_i)|L_{i,b}\rangle = -\lambda_i|\phi_i\rangle - (\lambda+1)W|\phi_j\rangle, \tag{C.3}
$$

$$
\langle\delta\phi_j|: (H-E_j)|L_{j,b}\rangle = -\lambda_j|\phi_j\rangle + \lambda W|\phi_i\rangle, \tag{C.4}
$$

$$
|\delta\phi_i\rangle: \langle L_{i,a}|(H-E_i) = -\lambda_i\langle\phi_i| + \lambda\langle\phi_j|W, \tag{C.5}
$$

and

$$
|\delta\phi_j\rangle: \langle L_{j,a}|(H-E_j) = -\lambda_j\langle\phi_j| - (\lambda+1)\langle\phi_i|W, \tag{C.6}
$$

which give $\lambda = -1/2$ and $\lambda_i = \lambda_j = -(\langle\phi_i|W|\phi_j\rangle)/2$. For the iterative approach, the function to optimize is

$$
M(L_{i/j}) = \langle L_{i/j}|(H_{mod,i/j}-E_{i/j})|L_{i/j}\rangle + 2\lambda_{i/j}\langle\phi_{i/j}|L_{i/j}\rangle + \langle\phi_{j/i}|W|L_{i/j}\rangle. \tag{C.7}
$$

For imaginary elements, we get

$$
\langle\delta\phi_i|: (H-E_i)|L_{i,b}\rangle = -\lambda_i|\phi_i\rangle - (\lambda+1)W|\phi_j\rangle, \tag{C.8}
$$

$$
\langle\delta\phi_j|: (H-E_j)|L_{j,b}\rangle = -\lambda_j|\phi_j\rangle - \lambda W|\phi_i\rangle, \tag{C.9}
$$

$$
|\delta\phi_i\rangle: \langle L_{i,a}|(H-E_i) = -\lambda_i\langle\phi_i| - \lambda\langle\phi_j|W, \tag{C.10}
$$

and

$$
|\delta\phi_j\rangle: \langle L_{j,a}|(H-E_j) = -\lambda_j\langle\phi_j| - (\lambda+1)\langle\phi_i|W, \tag{C.11}
$$

with $\lambda = -1/2$ and $\lambda_i = \lambda_j = -(\langle\phi_i|W|\phi_j\rangle)/2$.
For the iterative approach for the imaginary elements, one needs to optimize

$$
M(L_{i,b/j,b}) = \langle L_{i,b/j,b}|(H_{mod,i/j}-E_{i/j})|L_{i,b/j,b}\rangle + 2Re(\lambda_{i/j})\langle\phi_{i/j}|L_{i,b/j,b}\rangle. \tag{C.12}
$$

# D  Algorithm for one-qubit normalized states

We here describe in more detail the steps of the algorithm used for normalized states and a single qubit. In this example the Lagrange multipliers are computed iteratively. Given an Hamiltonian $H = \sum_i \gamma_i P_i$ with eigenbasis $|\phi_i\rangle$ and an Hermitian matrix $W = \sum_i \omega_i P_i$ that are linear combinations of Pauli strings, we want to find good, normalized, approximations $\phi_{i,t}$ and $\phi_{j,t}$ to the eigenstates of H, such that $\langle\phi_{i,t}|W|\phi_{j,t}\rangle$ is an element of W, through iterative calculation of Lagrange multipliers $|L_i\rangle$ and $|L_j\rangle$.

---

**Algorithm 1:** Algorithm for one-qubit with normalized states, iterative evaluation of Lagrangian, and real wave functions.

---

Initialize parameters $\theta_i, \theta_j, c_i, d_i, c_j, d_j$ ;
Compute the relevant overlaps;
Consider $W_{R/I} = \frac{W \pm W^T}{2}$;
**while** $\underline{F_\nu}$ not converged **do**

1. Compute $E_i = \left\langle \phi_{i,t} \middle| H \middle| \phi_{i,t} \right\rangle$, $H_{mod,i} = H - \frac{H|\phi_i\rangle\langle\phi_i|H}{\langle\phi_i|H|\phi_i\rangle}$ and set $\lambda = -\frac{1}{2}$

2. Derive the Lagrange multipliers from
   $M(|L_{i,\nu}\rangle) = \langle L_{i,\nu} | \left( H_{mod,i} - E_i \right) |L_{i,\nu}\rangle + \langle \phi_j | W_R | L_{i,\nu}\rangle$ for real elements and from
   $M(|L_{i,\nu}\rangle) = \langle L_{i,\nu} | \left( H_{mod,i} - \langle \phi_i | H | \phi_i \rangle \right) |L_{i,\nu}\rangle$ for imaginary ones, by solving the systems of derivatives with respect to $c_i/c_j$ and $d_i/d_j$ from Eq. 13

3. Find the partial derivatives of
   $F_\nu = \left\langle \phi_{i,t} \middle| W \middle| \phi_{j,t} \right\rangle + \left\langle L_{i,a} \middle| (H - E_i) \middle| \phi_{i,t} \right\rangle + \left\langle \phi_{i,t} \middle| (H - E_i)^\dagger \middle| L_{i,b} \right\rangle + \left\langle L_{j,a} \middle| (H - E_j) \middle| \phi_{j,t} \right\rangle$
   $+ \left\langle \phi_{j,t} \middle| (H - E_j)^\dagger \middle| L_{j,b} \right\rangle + \lambda \left[ \left\langle \phi_{i,t} \middle| W \middle| \phi_{j,t} \right\rangle \mp \left\langle \phi_{j,t} \middle| W \middle| \phi_{i,t} \right\rangle \right]$ with respect to $\theta_i$ and $\theta_j$ and solve the corresponding system of nonlinear equations to find the new parameters.

**end**

---

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
