# Peer review of "NISQ algorithm for the matrix elements of a generic observable"

_SciPost Physics, doi:SciPost Phys. 15, 180 (2023)_

## Round 1 · Referee Report · Anonymous (Referee 1) · 2023-1-10

Strengths

  1. Original Idea
  2. Potential for further work

Weaknesses

  1. Overstated claims
  2. Algorithmic procedure not clearly presented

Report

This article by Erbanni et al. introduces the idea of variational determining matrix elements of Hermitian operators in the eigenbasis of another hermitian operator - usually referred to as the Hamiltonian. The proposed concept is original, and applications of this idea are widely spread (e.g. in the fields of material science, chemistry and physics, as the article explains in its introduction), making it an interesting topic for the area of quantum computing/physics and the anticipated readers of scipost physics. Here I think a revised version could met the "Open a new pathway in an existing or a new research direction, with clear potential for multipronged follow-up work" acceptance criterium, as this type of non-direct variational optimization could inspire follow-up work.

Currently, the article appears to be premature with some of the claims and conclusions not justified by the presented data and analysis. I, therefore, can not recommend publication at this point. I anticipate however, that a revised version could be suitable for publication. Either reduced to a pure proof/show of concept with claims significantly redacted or with the original claims backed up by better data and more insightful analysis. In any case, the algorithmic part should be explained more clearly.

In the following, I will list my main concerns:

The conclusion that the approach is somehow resistant to randomized initialization can not be deduced from the presented analysis. First: Two individual one and two-qubit instances are not general enough to draw any such conclusion for general systems. Second: The two-qubit experiment is not randomly initialized but initialized with values close to the optimal angles. As the approach was not able to recover all matrix elements in this simple model system I would actually suspect that the approach is in general challenging to converge.

Claimed in introduction: "Various numerical simulations suggest that our approach manages to find many of the matrix elements even when one initializes randomly the trial functions over a very broad range of parameters"

Claimed in conclusion: "We have found that in general the method can perform well, meaning that it finds many of the matrix elements even when one initializes randomly the trial functions over a very broad range of parameters"

The overall presentation of the approach could be clearer and easier to follow. One suggestion would be to summarize it as an algorithmic procedure. Also the derivation and motivation could be improved (Eq. 1 for example kind of falls out of the sky).

In the approach, the trial states \phi_{i,t} and Multipliers \Lambda_{i,y} must be determined by fixing an ansatz and optimizing the angles. The straightforward variational approach would be to determine the eigenstates of the Hamiltonian E_i variationally and then compute <E_i|W|E_j> directly (e.g. via swap test). From a naive perspective this looks simpler as only half the number of states have to be determined, and in cases where the \phi_{i,t} are the same as E_i this would be the case. I assume this is not the case? This is hinted on a bit in the results section, the article would however benefit from a more general discussion as this is main difference to standard VQE procedures.

Some minor points:

The introduction claims that "off-diagonal matrix element calculation remains poorly understood" explained by "One reason for this is that an observable’s off-diagonal matrix elements can be a complex number, yet nearly all noisy intermediate scale quantum algorithms are designed to compute real values." Here I would disagree. Complex evaluation is a part of quite some NISQ procedures. For example, almost all approaches require an overlap estimation (e.g. Quantum Krylov or the Quantum-Assisted Simulation approach from one of the authors). I can also not see an inherent problem with the values being complex. I think the authors may want to state that one needs to be careful with complex entities for a variational approach, as one can not simply minimize them - one key aspect that makes the presented approach different from a standard VQE.

Fig.1 needs more information and a legend for the colors. Why are there 4 angles for the imaginary part but only 2 for the real part (degeneracies?).

Fig.2: description (dotted, dashed lines) does not match the figure (colours match, so the lines can still be identified correctly). The sentence: "The lack of a green dot-dashed line in panel (c) implies that this element did not appear in our attempts hybrid classicalquantum simulations with no error mitigation." is a bit unclear. Why did it not "appear"? Which value was computed instead?

Fig.2: Pannels are labelled as a-f but in the text they are referred to by the corresponding exact matrix elements for which the angles are computed. This is a bit exhausting to read.

"What is possibly more striking is that there are also converged results to values which do not belong to W2 , as for instance the value 30". Some analysis would be good here.

"scales polynomially with the system size (number of qubits) as this is often su"cient to obtain accurate results, e.g. using a Krylov basis [18]". Would consider citing some other krylov approaches than just the one of the co-author here (e.g. Stair/Evangelista, Kirby/Motta/Mezzacapo, Seki/Yunoki).

Comments on highlighted parts (blue):

  • Distinction to QSE and NO-VQE is well justified.
  • Information on overlap computation is sufficient.

Remarks that might be useful for the authors:

Why was the classical simulation done with individual shots? Hard to see the value in that. I would suspect that simulating exact overlaps gives a better picture of convergence in general. In the same manner, I wonder if noisy simulation in this setting gives meaningful insight. Doing noiseless (non-shot-based) simulations could be more helpful and save compute time.

---

## Round 1 · Referee Report · Anonymous (Referee 2) · 2023-1-23

Report

In this paper, the authors consider the question of how quantum computers, and particularly NISQ computers, may be used to determine the on- and off-diagonal matrix elements of operators. They discuss how it is an understudied question, particularly in light of the large amount of work put into NISQ algorithms for various properties (such as ground state estimation and Hamiltonian simulation). Obtaining the off-diagonal matrix elements of operators is an important task for many subfields of chemistry and physics. Even more works are done on the diagonal matrix elements. I was surprised that the introduction does not mention any of the grouping or classical shadow tomography techniques that were developed recently. Even if some of these works were for diagonal matrix elements they can be easily repurposed for off-diagonal elements using Ref 10 equation (15), which adds one extra qubit. The entire introduction gives an impression that the authors are not aware of modern developments in quantum measurement problem. As for their method, it is not motivated at all, why would anyone need Lagrange multiplier method to evaluate matrix elements? This is the question that the work should address instead of telling how it is done, it is always imperative to answer the why question. The exposition is not clear because the explanation is very poorly done. Results do not present clear advantage compare to other methods (no other methods were presented): the basic standard in the literature is to present the number of measurements needed to achieve a certain accuracy (e.g. milli-Hartree). None of this is done here so it is hard to judge whether this approach is any better than previously reported ones.

Considering all these problems, I do not recommend the publication.

Here are some references on previous measurement methods developed in the field recently:

Phys. Rev. X 10, 031064 (2020)
npj Quantum Inf 7, 23 (2021)
Phys. Rev. Lett. 127, 030503 (2021)
Quantum 5, 385 (2021)
PRX Quantum 2, 040320 (2021)
J. Chem. Theory Comput. 18, 7394 (2022)
Commun. Math. Phys. 391, 951–967 (2022)
Quantum 7, 889 (2023)

---

## Round 1 · Author Response

Dear Editor,

Thank you for handling our submission. In your reply to our submission, you asked
for an explanation of how to efficiently compute the overlaps in Eq. 1-6 on a quantum
computer and for a comparison with the quantum subspace expansion in Ref.[8].
Please find attached the resubmitted version, where we have addressed the issues you
raised. In particular, the forms we consider for the trial wavefunctions and the trial
Lagrange multipliers depend on a polynomial number of terms, and likewise, we
consider matrices H and W that are linear combinations of a polynomial number of
k-local unitaries, i.e. that act non-trivially on at most k qubits which were shown in
Ref.[51] to allow for the efficient computation of their expectation values.
Theoretically, since our method is iterative, we would need new evaluations of each of
these overlaps for each iteration of the loop. In practice, though, given the forms of our
trial quantities in Eq.11-13, we just need to compute the estimates of the elements of
H and W in the computational basis once, before the start of the optimization routine,
during which we just multiply them by their respective coefficients, which come as a
result of the optimization process.
Last, in Ref.[8], the authors focus on the ground and excited states, while our approach
also allows estimating the off-diagonal elements of W.
We thus hope that this new version can be considered for publication in SciPost.

Kind regards,
Authors

---

## Round 1 · List of Changes

1. Added the following paragraph in the updated manuscript: "While our choice of ansatz may seem to resemble existing works in the literature [8, 12, 18], none of these results works for the off-diagonal matrix elements of a generic observable. Moreover, our approach is fundamentally different and uses Lagrange multipliers to encode the constraints for the underlying problem into the refined objective."

  2. Added a section on scaling analysis for the overlap computation

---

## Round 2 · Referee Report · Anonymous (Referee 3) · 2023-4-4

Strengths

1.- Novel attempt to use variational quantum algorithms beyond ground-state energies.

Weaknesses

1.- No description of the sampling complexity of the algorithm, making it difficult for a reader to assess the its practical implementation.
2.- Claim of scalable method solely based on a one and two qubit numerical experiment with the two-qubit experiment is already poorly performing.
3.- Main goal of the work is difficult to extract from the abstract and introduction.
4.- Poor motivation of the work.

Report

Summary: The manuscript "NISQ algorithm for the matrix elements of a generic observable" describes a method to calculate the matrix elements of a generic quantum observable from the eigenstates of a Hamiltonian (e.g. in the energy bases). The main result the use of Lagragian multiplyers to extract diagonal and off-diagonal elements of a quantum observable by optimizing the parameters of constrained quantum states.

Decision: I am unable to accept the manuscript for publication in SciPost. The main reason for this decision is the impossibility to assess if the algorithm will be practical for even moderate size problems. The sampling complexity analysis is almost inexistent, and a reader has a hard time knowing how many times a quantum computer needs to be called for even N=1,2 qubits. On the same line going from 1 to 2 qubit in a simulation is not sufficient to claim "scalability", moreover when the 2 qubit experiment already show poor performance.

Despite the decision I acknowledge the novelty of the work as an attempt to use variational quantum algorithms beyond ground-state energies. I, therefore encourage the authors to further work on the idea presented in this manuscript and mature the results.

Comments: - The manuscript is overall poorly written. - The main goal is hard to extract from either the abstract or the introduction. - It is unclear why someone should care about calculating off-diagonal elements of an operator in a known basis. - The complexity of the algorithm (in terms of the number of queries to a quantum device) is impossible to extract from the manuscript. This result is critical to assess whether the protocol is practical. I suggest to pick an example with known scaling and count the queries required to implement the algorithm. - From one and two qubit simulations is not possible to draw any conclusion on the scalability of the algorithm.

  • validity: low
  • significance: ok
  • originality: good
  • clarity: poor
  • formatting: below threshold
  • grammar: below threshold

---

## Round 2 · Referee Report · Anonymous (Referee 1) · 2023-5-15

Report

As stated in the initial report, the proposed concept is original, and applications of this idea are widely spread. The exaggerated claims made in the original manuscript have been appropriately toned down, resulting in a satisfactory level that now accurately portrays the developed methodologies and numerical demonstrations in an honest manner.

Survey of the literature with similar content was improved as well and is in this form acceptable. I appreciate the effort that went into the changes on page 4 where the difference are clearly worked out. This will be beneficial to a lot of readers, especially students.

I recommend publication.

Thanks for clearly marking changes in the manuscript & apologies for the delay in my report.

Minor comment:

The newly added clarification before Eq (1) is good, but might be confusing to some on first read as: Re(M) = 0.5( M + M^T) and similar for the imaginary part only holds for M Hermitian and not for general matrices.
This is implicitly given by stating that W is an observable further above, I would however change "We first note that, for a given matrix W, we can al-
ways write" to "We first note that, for a given Hermitian matrix W, we can al-
ways write". Or replace "T" with "*" and clarify, that for Hermitian matrices W^T = W^*

---

## Round 2 · Author Response

Our reply contains two new plots, so we uploaded it along with the main file in the zip folder attached to this resubmission.

---

## Round 2 · List of Changes

We highlighted the changes in blue in both the main pdf and the resubmission letter.

---

## Round 3 · Referee Report · Anonymous (Referee 5) · 2023-8-3

Strengths

1.- Improved writing and readability.

Weaknesses

1.- The scaling calculation only concerns the number of measurements for the overlaps, but still lacks on the sampling complexity, and how it affects the algorithm.
2.- Only 2 qubit experiments with poor performance to claim scalability, unacceptable.

Report

I am unable to accept this paper for publication. I acknowledge the improvements on the manuscript but the work is still not mature enough to be published.
My main reasons are based ont he following points:
- Claiming scalability based on numerical evidence with only one and two qubit experiments is not enough.
- The scaling analysis only concerns one part of the algorithm, but ignores the rest.

---

## Round 3 · Referee Report · Anonymous (Referee 6) · 2023-8-9

Strengths

1. High/Improved readability
2. Main Ideas explained with sufficient detail
3. Sufficient comparisons to related methods
4. Data is presented nicely
5. Calculations on the simple demonstrations are quite extensive
6. No overstated claims anymore
7. Almost all points of previous reports addressed

Weaknesses

1. Only 1 and 2 qubit demonstrations (remains a weakness)
2. Scalability unclear (analysis improved though)

Report

The current version of the article presents a conceptually intriguing idea in a well-organized manner. The primary focus revolves around the estimation of matrix elements for general observables within the eigenbasis of a specified Hamiltonian. Unlike the conventional approach, which entails solving for eigenstates followed by matrix element computation, the proposed methodology directly constructs a corresponding objective function.

Although approaches like this exist in the literature (as research on variational optimization has always been quite rich) these concepts might remain unfamiliar to a substantial portion of the quantum computing community. The present work is primarily dedicated to elucidating the key components for constructing suitable objective functions.

The concept is illustrated on a single and two-qubit example. Admittedly, these instances lean towards extreme simplicity, functioning more as preliminary demonstrations rather than serving as definitive numerical proofs of the approach's applicability. The authors do also not claim such things (in this revised version).

Acknowledging the reservations expressed by colleagues in other reports, I share their skepticism towards deriving meaningful numerical insights from experiments involving just 1-2 qubits. I think however, that in the present case, it is more forgivable than in most other works with similar shortcomings. What speaks for the presented data beyond a mere didactical demonstration is that potential obstacles can already be detected at the two-qubit level, indicating that more work needs to be done to make the approach practicable. One can for example already see effects of (simulated) device-noise and shot-noise.

Fair comparison to alternative approaches (e.g. sequential VQEs + measurement matrix element) are at this stage out of scope for this work, as there is a high dependence on a number of different parameters intrinsic to the involved methologies (choice of test systems, choice of VQE ansatz, choice of the ansätze for the methodology introduced here, .... and many more). The authors choice to not do further numerics in this direction but rather discuss the differences to other prominent methods at high-level in the text is therefore understandable.

Requested changes

From context I assume that qiskit was used to simulate the results ("IBM QPU simulator"). It should be cited accordingly.

---

## Round 3 · Author Response

Our reply contains again two new plots, so we uploaded it along with the main file in the zip folder attached to this resubmission.

---

## Round 3 · List of Changes

We highlighted the changes in blue in the main pdf and the resubmission letter.

---

## Editorial Decision

published